Evolution of dopamine receptors: phylogenetic evidence suggests a later origin of the DRD2l and DRD4rs dopamine receptor gene lineages

Opazo Juan C. jopazo@gmail.com 1
Zavala Kattina 1
Miranda-Rottmann Soledad 2
Araya Roberto 2
1 Instituto de Ciencias Ambientales y Evolutivas, Universidad Austral de Chile , Valdivia , Chile
2 Department of Neurosciences, Faculty of Medicine, University of Montreal , Montreal , Canada
Wilke Claus
Electronic publication date: 2018 Apr 13
Publication date: 2018
Volume: 6
Electronic Location ID: e4593
Received 2018 Jan 11; Accepted 2018 Mar 17
Copyright: ©2018 Opazo et al.
Copyright year: 2018
Copyright holder: Opazo et al.
License: This is an open access article distributed under the terms of the Creative Commons Attribution License, which permits unrestricted use, distribution, reproduction and adaptation in any medium and for any purpose provided that it is properly attributed. For attribution, the original author(s), title, publication source (PeerJ) and either DOI or URL of the article must be cited.
License URL: https://creativecommons.org/licenses/by/4.0/

Keywords: Dopamine receptors, Gene family evolution, Whole genome duplications, Neuroscience

Funding: Fondo Nacional de Desarrollo Científico y Tecnológico (FONDECYT) 1160627 Canadian Institutes of Health Research (CIHR) MOP 133711 Natural Sciences and Engineering Research Council of Canada (NSERC Dicovery Grant) 418113-2012 (NSERC PIN 392027) This work was supported by the Fondo Nacional de Desarrollo Científico y Tecnológico (FONDECYT) grant 1160627 to Juan C. Opazo, by the Canadian Institutes of Health Research (CIHR) grant MOP 133711 and by the Natural Sciences and Engineering Research Council of Canada (NSERC Dicovery Grant) grant application No. 418113-2012 (NSERC PIN 392027) to Roberto Araya. The funders had no role in study design, data collection and analysis, decision to publish, or preparation of the manuscript.

==============================
Dopamine receptors are integral membrane proteins whose endogenous ligand is dopamine. They play a fundamental role in the central nervous system and dysfunction of dopaminergic neurotransmission is responsible for the generation of a variety of neuropsychiatric disorders. From an evolutionary standpoint, phylogenetic relationships among the DRD1 class of dopamine receptors are still a matter of debate as in the literature different tree topologies have been proposed. In contrast, phylogenetic relationships among the DRD2 group of receptors are well understood. Understanding the time of origin of the different dopamine receptors is also an issue that needs further study, especially for the genes that have restricted phyletic distributions (e.g., DRD2l and DRD4rs). Thus, the goal of this study was to investigate the evolution of dopamine receptors, with emphasis on shedding light on the phylogenetic relationships among the D1 class of dopamine receptors and the time of origin of the DRD2l and DRD4rs gene lineages. Our results recovered the monophyly of the two groups of dopamine receptors. Within the DRD1 group the monophyly of each paralog was recovered with strong support, and phylogenetic relationships among them were well resolved. Within the DRD1 class of dopamine receptors we recovered the sister group relationship between the DRD1C and DRD1E, and this clade was recovered sister to a cyclostome sequence. The DRD1 clade was recovered sister to the aforementioned clade, and the group containing DRD5 receptors was sister to all other DRD1 paralogs. In agreement with the literature, among the DRD2 class of receptors, DRD2 was recovered sister to DRD3, whereas DRD4 was sister to the DRD2/DRD3 clade. According to our phylogenetic tree, the DRD2l and DRD4rs gene lineages would have originated in the ancestor of gnathostomes between 615 and 473 mya. Conservation of sequences required for dopaminergic neurotransmission and small changes in regulatory regions suggest a functional refinement of the dopaminergic pathways along evolution.

Introduction

The availability of whole genome sequences offers a great opportunity to study the evolution of genes involved in physiological processes in a variety of living organisms. The diversity of gene content and its evolutionary history are fundamental pieces of information that should be taken into account when comparing the physiology of different species. To understand the evolution of genes it is necessary to reconcile their evolutionary history by comparing relationships among genes—i.e., gene trees—and among species involved in the study—i.e., species trees. Thus, comparing both trees represents a powerful approach to infer homology, time of origin, birth-and-death processes, gene conversion events among others.

Dopamine receptors are integral membrane proteins that mediate the action of dopamine (Beaulieu & Gainetdinov, 2011). They play fundamental roles in functions associated with the central nervous system including learning, cognition, memory, feeding, sleep, and motor control among others (Beaulieu & Gainetdinov, 2011). Peripherally, these receptors are also involved in hormonal regulation, cardiovascular function, renal function, and olfaction among others (Beaulieu & Gainetdinov, 2011). Several human disorders are associated with dopamine receptors including parkinson’s disease, schizophrenia, Tourette’s syndrome, Huntington’s disease, drug abuse and addiction, bipolar disorder, depression, and hypertension among others (Hussain & Lokhandwala, 1998; Hisahara & Shimohama, 2011; Chu et al., 2012; Chen et al., 2013; Denys et al., 2013; Brisch, 2014; Ashok et al., 2017). Based on their pharmacological properties, dopamine receptors are classified into two major groups: the DRD1 group, which includes DRD1, DRD5, DRD1C, and DRD1E; and the DRD2 group that includes DRD2, DRD2l, DRD3 DRD4, and DRD4rs (Yamamoto et al., 2015). Today it is well known that these groups originated independently such that the ability to bind dopamine was acquired twice during the evolution of biogenic amine receptors (Callier et al., 2003; Yamamoto et al., 2013; Yamamoto et al., 2015; Spielman, Kumar & Wilke, 2015). Although both groups share the ability to bind dopamine, they also show the signature of their independent histories as they differ in several other characteristics (Sibley, 1999; Beaulieu & Gainetdinov, 2011). From a structural standpoint, the DRD1 class of receptors is characterized by the lack of introns, a short third cytoplasmatic loop, and a long C-terminal tail. Conversely, DRD2 possess up to six introns, encoding a long third cytoplasmatic loop and a short C-terminal tail (Gingrich & Caron, 1993). From a biochemical perspective, the DRD1 group of receptors activates the GαS∕olf family of G proteins stimulating adenilate cyclase activity and production of CAMP. The DRD2 group of receptors, on the other hand, activates the Gαi∕o family of G proteins inhibiting adenilate cyclase activity and reducing levels of CAMP (Sibley, 1999; Beaulieu & Gainetdinov, 2011). Regarding the synaptic anatomy, the DRD1 class of receptors is located exclusively at the postsynaptic site whereas the DRD2 class is found in both pre- and postsynaptic terminals (Sibley, 1999; Beaulieu & Gainetdinov, 2011; Araya et al., 2013).

From an evolutionary standpoint, evolutionary relationships among the members of the DRD1 class of dopamine receptors are still a matter of debate; different phylogenetic hypotheses have been proposed in the literature. For example, DRD1 has been recovered sister to DRD5, a clade that in turn is recovered sister to DRD1C; in these studies DRD1E is recovered sister to all other DRD1 members (Callier et al., 2003; Yamamoto et al., 2013). In other cases, the clade containing DRD1 sequences has been recovered sister to DRD1C, and this group is sister to DRD5 (Le Crom et al., 2004). A case in which the monophyly of DRD1E has not been recovered has also been reported (Haug-Baltzell et al., 2015). There is also a case in which the members of the DRD1 class of dopamine receptors have been recovered as two distinct clades, one that includes DRD1 and DRD5 and another grouping DRD1C and DRD1E (Yamamoto et al., 2015). In contrast to the lack of phylogenetic agreement among the DRD1 class of dopamine receptors, phylogenetic relationships among the members of the DRD2 class of dopamine receptors are well resolved as in most studies DRD2 is recovered sister to DRD3, whereas DRD4 is recovered sister to the DRD2/DRD3 clade (Callier et al., 2003; Haug-Baltzell et al., 2015; Spielman, Kumar & Wilke, 2015; Yamamoto et al., 2015). Understanding the time of origin of the different dopamine receptors is also an issue that needs further study, especially for the genes that possess restricted phyletic distributions (e.g.,  DRD2l and DRD4rs). Regarding the time of origin, different hypotheses are associated with different phylogenetic predictions. Therefore, a phylogenetic tree that is built on adequate taxonomic sampling and an adequate number of genes should provide valuable information to understand the time of origin of dopamine receptors and also about their sister group relationships.

The goal of this study was to investigate the evolution of dopamine receptors, with emphasis on shedding light on the phylogenetic relationships among the DRD1 class of dopamine receptors and the time of origin of the DRD2l and DRD4rs gene lineages. Our results recovered the monophyly of the two groups of dopamine receptors. Within the DRD1 class of receptors, the monophyly of each paralog was recovered with strong support, and phylogenetic relationships among them were well resolved. We recovered the sister group relationship between the DRD1C and DRD1E receptors, and this clade was recovered sister to a cyclostome sequence. The DRD1 clade was recovered sister to the aforementioned clade, and the group containing the DRD5 receptors was sister to all other DRD1 paralogs. This topology represents a new phylogenetic hypothesis for the evolution of this group of dopamine receptors. In agreement with the literature, among the DRD2 class of dopamine receptors, DRD2 was recovered sister to DRD3 whereas DRD4 was sister to the DRD2/DRD3 clade. Finally, our phylogenetic evidence suggests a later origin of the DRD2l and DRD4rs gene lineages.

Materials and Methods

DNA data and phylogenetic analyses

We used bioinformatic procedures to retrieve dopamine receptor genes in species of all major groups of vertebrates. Our sampling included mammals, birds, reptiles, amphibians, coelacanths, teleost fish, holostean fish, cartilaginous fish and cyclostomes (Table S1). We identified genomic pieces containing dopamine receptor genes in the Ensembl database using BLASTN with default settings (Maximum number of hits to report =100; Maximum E-value for reported alignments =10; Word size for seeding alignments =11; Match/Mismatch scores =1,  − 3; Gap penalties: opening =5, Extension =2) or NCBI database (refseq_genomes, htgs, and wgs) using tbalstn (Altschul et al., 1990) with default settings (Max target sequences =100; Expect threshold =10; word size =6; Max matches in a query range =0; Matrix = BLOSUM62; Gap cost: Existence =11, Extension =1; Compositional adjustments: Conditional compositional score matrix adjustment) (human and chicken sequences were used as a starting point). Conserved synteny, based on previous literature (Yamamoto et al., 2013; Yamamoto et al., 2015), was also used as a criterion to define the genomic region containing dopamine receptor genes. Once identified, genomic pieces were extracted including the 5′ and 3′ flanking genes. After extraction, we curated the existing annotation by comparing known exon sequences to genomic pieces using the program Blast2seq with default parameters (Max target sequences =100; Expect threshold =10; word size =28; Max matches in a query range =0; Match/Mismatch scores: 1,  − 2; Gap costs = Linear) (Tatusova & Madden, 1999). Putatively functional genes were characterized by an open intact reading frame with the canonical exon/intron structure typical of vertebrate dopamine receptors. Sequences derived from shorter records based on genomic DNA or cDNA were also included in order to attain a broad and balanced taxonomic coverage. We also included sequences of the α2-adrenoreceptors (ADRA2A, ADRA2B, ADRA2C, ADRA2D), and β-adrenoreceptors (ADRB1, ADRB2 and ADRB3) (Table S1). Our final dataset contained 396 sequences. Amino acid sequences were aligned using the FFT-NS-i strategy from MAFFT v.7 (Katoh & Standley, 2013). We used the proposed model tool of IQ-Tree (Trifinopoulos et al., 2016) to select the best-fitting model of amino acid substitution (JTT + R9). We performed a maximum likelihood analysis to obtain the best tree using the program IQ-Tree (Trifinopoulos et al., 2016); support for the nodes was assessed with 1,000 bootstrap pseudoreplicates using the ultrafast routine. Phylogenetic analyses were performed 20 times in order to better explore the tree space. Human ADRA1A, ADRA1B, and ADRA1D sequences were used as outgroups.

Assessments of conserved synteny

We examined genes found upstream and downstream of the dopamine receptor genes of representative vertebrate species. We used the estimates of orthology and paralogy derived from the EnsemblCompara database (Herrero et al., 2016); these estimates are obtained from an automated pipeline that considers both synteny and phylogeny to generate orthology mappings. These predictions were visualized using the program Genomicus v90.01 (Louis et al., 2015). Our assessments were performed in humans (Homo sapiens), chicken (Gallus gallus), spotted gar (Lepisosteus oculatus) and elephant shark (Callorhinchus milii). In the case of the elephant shark, flanking genes were examined using the entrez gene database from the National Center for Biotechnology Information (NCBI) (Maglott et al., 2011).

Molecular structure and graphics

Molecular visualization and analyses of the human DRD4 protein structure were performed with the UCSF Chimera package (Pettersen et al., 2004) using the 1.96 Å resolution structural file PDB ID: 5WIV (Wang et al., 2017). Molecular dynamics simulation of site-directed mutagenesis was performed using the Chimera structure editing tool and choosing the Dunbrack rotamer library (Dunbrack, 2002) to visualize the probability of a particular amino acidic conformation. The rotamer displaying the highest probability was selected: Y rotamer : 72.6% probability; I rotamer 79% probability. Sequences were aligned using Vector NTI Express (Thermo Fisher) using default parameters (Display setup: identity value = 1, Similarity value = 0.5, Weak similarities value = 0.2. Showing weak similarities = checked. Multiple alignment options: slow, Protein weight matrix = GONNET, gap open penalty = 15, gap extension penalty = 6.66, percentage of identity for delay = 30. Protein gap parameters: hydrophilic residues = GPSNDQEKR, Gap separation distance = 4, residue specific penalties = checked, hydrophilic penalties = checked). Human protein sequences DRD2: NP_000786.1 and DRD4: NP_000788 were used as reference for the numbering and alignment.

Results and Discussion

Overview of the evolution of dopamine receptors

In this work we performed an evolutionary study of dopamine receptors in representative species of all major groups of vertebrates. We combined gene phylogenies and synteny analyses with the main goal of understanding the duplicative history of the DRD1 class of dopamine receptors and the time of origin of the DRD2l and DRD4rs gene lineages. Our phylogenetic tree recovered the monophyly of the two groups of dopamine receptors (Fig. 1). In the first clade we recovered the sister group relationship between the DRD1 class of receptors and a clade containing β-adrenoreceptors (Fig. 1); in the second clade, the DRD2 receptors were recovered sister to the α2-adrenoreceptors (Fig. 1). This phylogenetic arrangement is in agreement with previous results (Yamamoto et al., 2013; Spielman, Kumar & Wilke, 2015; Céspedes et al., 2017; Zavala et al., 2017) and reflects the fact that the ability to bind dopamine was acquired twice during the evolutionary history of biogenic amine receptors (Callier et al., 2003; Yamamoto et al., 2015).

Figure 1 Maximum likelihood tree depicting evolutionary relationships among dopamine receptors in vertebrates.

Numbers on the nodes correspond to maximum likelihood ultrafast bootstrap support values. Human ADRA1A, ADRA1B, and ADRA1D sequences were used as outgroups.

Phylogenetic relationships among the DRD1 class of dopamine receptors

According to our phylogenetic analyses, the monophyly of the DRD1 class of dopamine receptors, as well as the monophyly of each paralog (DRD1, DRD5, DRD1C and DRD1E), were recovered with strong support (Fig. 1). In all cases synteny analyses provided further support for the identity of the four DRD1 clades recovered in our phylogenetic tree (Fig. 2). Phylogenetic relationships among the different DRD1 lineages were well resolved (Fig. 1). We recovered the sister group relationship between the DRD1C and DRD1E dopamine receptors (Fig. 1), and this clade was recovered sister to a cyclostome sequence (Fig. 1). The DRD1 clade was recovered sister to the aforementioned clade, and the group containing DRD5 sequences was recovered sister to all other DRD1 paralogs (Fig. 1). Although in the literature there are studies reporting dopamine receptor phylogenies (Callier et al., 2003; Le Crom et al., 2004; Yamamoto et al., 2013; Yamamoto et al., 2015; Haug-Baltzell et al., 2015), they are not directly comparable as the taxonomic and/or family membership sampling differ. Beyond this point, phylogenetic relationships among the DRD1 class of dopamine receptors seem to still be a matter of debate. In some cases DRD1 has been recovered sister to DRD5, a clade that in turn is recovered sister to DRD1C; in these studies DRD1E is recovered sister to all other DRD1 (Callier et al., 2003; Le Crom et al., 2003; Yamamoto et al., 2013). In other studies the clade containing DRD1 sequences has been recovered sister to DRD1C, and this group is sister to DRD5 (Le Crom et al., 2004). A case in which the monophyly of DRD1E is not recovered has also been reported (Haug-Baltzell et al., 2015). Finally, there is also a case in which the DRD1 class of receptors has been recovered as two different clades, one that includes DRD1 and DRD5 and another grouping DRD1C and DRD1E (Yamamoto et al., 2015). Thus, our results propose a new phylogenetic hypothesis regarding the evolution of the DRD1 class of dopamine receptors (Fig. 1). Overall, we believe that our hypothesis is well supported based on a taxonomic sampling that covered all main groups of vertebrates, as well as, the phylogenetic context of the monoamine receptors (Spielman, Kumar & Wilke, 2015).

Figure 2 Patterns of conserved synteny in the chromosomal regions that harbor the DRD1 class of dopamine receptors.

(A) Chromosomal region that harbors the DRD1 gene; (B) Chromosomal region that harbors the DRD1C gene; (C) Chromosomal region that harbors the DRD1E gene; (D) Chromosomal region that harbors the DRD5 gene. Asterisks denote that the orientation of the genomic piece is from 3′ to 5′, gray lines represent intervening genes that do not contribute to conserved synteny whereas dashed lines represent genes that are not present.

Phylogenetic relationships among the DRD2 class of dopamine receptors

We recovered the monophyly of the DRD2 class of dopamine receptors with strong support (Fig. 1). The monophyly of all paralogs of this class of receptors are also well supported, defining clear orthology and paralogy (Fig. 1). Synteny analyses provide further support for the evolutionary identity of all DRD2 dopamine receptors (Fig. 3). In our phylogenetic tree DRD2 was recovered sister to DRD3 with strong support (Fig. 1), whereas DRD4 was sister to the DRD2/DRD3 clade (Fig. 1). In contrast to the lack of phylogenetic resolution among the DRD1 class of dopamine receptors, phylogenetic relationships among the DRD2 class of receptors seem to be well resolved as all studies, including ours, show the same topology ((DRD2,DRD3),DRD4) (Callier et al., 2003; Le Crom et al., 2003; Haug-Baltzell et al., 2015; Spielman, Kumar & Wilke, 2015; Yamamoto et al., 2015).

Figure 3 Patterns of conserved synteny in the chromosomal regions that harbor the DRD2 class of dopamine receptors.

(A) Chromosomal region that harbors DRD2 gene; (B) Chromosomal region that harbors DRD21 gene; (C) Chromosomal region that harbors DRD3 gene; (D) Chromosomal region that harbors DRD4 gene; (E) Chromosomal region that harbors DRD4rs gene. Asterisks denote that the orientation of the genomic piece is from 3′ to 5′, gray lines represent intervening genes that do not contribute to conserved synteny whereas dashed lines represent genes that are not present.

Phylogenetic evidence for the origin of the DRD2l gene lineage in the ancestor of gnathostomes

In agreement with Yamamoto et al. (2015), our phylogenetic analyses also suggest the presence of an extra dopamine receptor gene lineage that is related to DRD2 gene (Boehmier et al., 2004; Boehmler et al., 2007) (Fig. 4). Although our results agree with Yamamoto et al. (2015) regarding the presence of a new dopamine receptor gene lineage, our results suggest a different time of origin.

Figure 4 Maximum likelihood trees depicting evolutionary relationships among DRD2 and DRD2l dopamine receptors in vertebrates.

Numbers on the nodes correspond to maximum likelihood ultrafast bootstrap support values. This tree topology does not represent novel phylogenetic analyses; they are the DRD2/DRD2l clade that was recovered from Fig. 1.

According to our results, we recovered a strongly supported clade containing the DRD2l sequences of teleost fish, holostean fish, and coelacanths (Fig. 4) sister to the clade containing DRD2 sequences of gnathostomes (Fig. 4). This tree topology suggests that in the ancestor of gnathostomes, between 615 and 473 mya, the DRD2 gene underwent a duplication event that gave rise to an extra DRD2 gene copy—the DRD2l—that was independently lost in the ancestor of tetrapods and cartilaginous fish (Fig. 5). In support of this scenario, our phylogenetic tree recovered a cyclostome sequence sister to the DRD2/DRD2l clade (Fig. 4). The pattern of gene conservation found up, and downstream of DRD2 and DRD2l genes, provides further support for the presence of two DRD2 dopamine receptor gene lineages (Fig. 3). For example, in the spotted gar (Lepisosteus oculatus), a species that possesses both DRD2 gene copies, DRD2 and DRD2l are found in different chromosomal locations. The identity of their genomic locations is defined by the presence of upstream and downstream flanking genes all across gnathostome vertebrates. Thus, the upstream genes ANKK1 and TTC12 and the downstream genes TMPRSS, ZW10, USP28 and HTR3B define the genomic location of the DRD2 gene lineage, whereas the upstream gene XRCC1 and downstream genes ETHE1, PHLDB3 and IRQQ1 define the genomic location of the DRD2l gene lineage (Fig. 3). Importantly, this pattern of conservation is also found in species that lost the DRD2l gene from their genomes (Fig. 3).

Figure 5 Phyletic distribution of dopamine receptor genes in vertebrates.

The cyclostome check between the DRD1C and DRDE indicates that the duplication event that gave rise to these genes occurred after the divergence between cyclostomes and gnathostomes. Therefore, cyclostomes retained the ancestral condition of a single gene copy. A similar situation applies to DRD2/DRD2l and to DRD4/DRD4rs.

The evolutionary hypothesis proposed here is different from that proposed by Yamamoto et al. (2015) in which the clade containing DRD2l sequences was recovered sister to a clade containing DRD2 sequences of vertebrates. Thus, according to their phylogeny the duplication event that gave rise to the DRD2l gene would have occurred in the ancestor of vertebrates, between 676 and 615 mya, even though they claim that the origin of this gene occurred after the Osteichthyes-Chondrichthyes divergence, between 473 and 435 mya (Yamamoto et al., 2015). Beyond this discrepancy, both evolutionary scenarios proposed in the study of Yamamoto et al. (2015) are different from ours.

Figure 6 Alignment of the human dopamine receptor 2 (DRD2) with zebrafish (Danio rerio), coelacanth (Latimeria chalumnae) and spotted gar (Lepisosteus oculatus) dopamine receptor 2l (DRD2l).

Shaded regions denote transmembrane domains according to UniProt. Dopamine binding sites, agonist and antagonist binding sites were predicted with theoretical and computational techniques (Yashar et al., 2004) and experimental evidence (Shi & Javitch, 2002) . Amino acids in the third intracellular loop conferring G protein subunit Gαi specificity (Senogles et al., 2004) are indicated by orange asterisks.

An amino acid alignment of both DRD2 gene lineages revealed that in the case of the spotted gar (Lepisosteus oculatus) and the coelacanth (Latimeria chalumnae) the distance, defined as the percentage of amino acid residues that are different between two sequences, between DRD2 and DRD2l receptors is approximately 30% whereas it is approximately 45% in zebrafish (Danio rerio). These estimates are in agreement with previous reports (Boehmier et al., 2004). Additionally, the human DRD2 amino acid sequence was aligned to the zebrafish, coelacanth and spotted gar DRD2l sequence to infer functionally significant changes (Fig. 6). The binding sites for dopamine and DRD2 agonists and antagonists are conserved among these species. However, the adjacent hydrophobic pocket, which confers ligand specificity to DRD2 is not conserved (Fig. 6). While in humans, coelacanths and spotted gar the second amino acid of the third transmembrane domain (TM3) is phenylalanine (F), it is leucine (L) in zebrafish. This change from an aromatic to an aliphatic amino acid could change the zebrafish DRD2l ligand specificity and therefore its function. The site that confers specificity to the human G protein subunit Gαi (Senogles et al., 2004) (Fig. 6; orange asterisks) is not conserved among species. The side chain size, shape and polarity changes observed could potentially influence the receptor/G protein coupling specificity, suggesting important evolutionary differences.

Phylogenetic evidence for the origin of DRD4rs gene lineage in the ancestor of gnathostomes

Also in agreement with Yamamoto et al. (2015) our phylogenetic reconstruction identified an extra dopamine receptor gene lineage that is related to the DRD4 gene (Figs. 1 and 7). According to our phylogenetic tree, a strongly supported clade that contains dopamine receptors of bony fish and coelacanths was recovered sister to the DRD4 clade of gnathostomes (Fig. 7). Similarly to the DRD2l gene lineage, this topology suggests that the DRD4 gene underwent a duplication event in the ancestor of gnathostomes, between 615 and 473 mya, giving rise to an extra copy of the DRD4 gene. During the radiation of the group, one of the copies (DRD4) was retained in all main groups of gnathostomes, whereas the other was only retained in bony fish and coelacanths (Fig. 5). In agreement with this hypothesis, our phylogenetic reconstruction recovered a lamprey sequence sister to the DRD4/DRD4rs clade (Fig. 7). Synteny analyses provide further support to our phylogenetic tree, as the genomic locations that harbor both DRD4 gene lineages are different (Fig. 3). Thus, there are four upstream genes (SCT, CDHR5, IRF7 and PHRF1) and four genes downstream (DEAF1, TMEM80, EPS8L2 and TALDO1) that define the identity of the DRD4 genomic location (Fig. 3). Similarly, there are upstream genes (KCP, CDHR5, IRF5 and TNOP3) and downstream genes (ATP6V1F) of the DRD4rs gene that define the identity of its genomic location (Fig. 3). Similar to that found for the DRD2 genes, our evolutionary hypothesis regarding the origin of the DRD4rs gene lineage is different from the scenario proposed by Yamamoto et al. (2015). According to their results, the clade containing DRD4rs sequences was recovered sister to a clade containing DRD4 sequences of vertebrates. Therefore, their phylogenetic tree suggests that the evolutionary origin of the DRD4rs gene lineage would be in the ancestor of vertebrates, between 676 and 615 mya, as a product of two rounds of whole genome duplication (Yamamoto et al., 2015). Thus, both studies suggest different evolutionary scenarios regarding the time of origin of the DRD4rs gene lineage.

Figure 7 Maximum likelihood trees depicting evolutionary relationships among DRD4 and DRD4rs dopamine receptors in vertebrates.

Numbers on the nodes correspond to maximum likelihood ultrafast bootstrap support values. This tree topology does not represent novel phylogenetic analyses; they are the DRD4/DRD4rs clade that was recovered from Fig. 1.

Figure 8 Alignment of the human dopamine receptor 4 (DRD4) with zebrafish (Danio rerio), coelacanth (Latimeria chalumnae) and spotted gar (Lepisosteus oculatus) dopamine receptor 4rs (DRD4rs).

Shaded regions denote transmembrane domains according to UniProt. Dopamine binding sites (red dots) were determined by site directed mutagenesis (Cummings et al., 2010) and homology to DRD2. Antagonist binding sites and hydrophobic pocket-including selectivity region-were obtained from mutagenesis studies (Cummings et al., 2010) and from the crystal structure of the receptor coupled to the antagonist nemonapride (Wang et al., 2017). Non-conserved amino acids in the nemonapride binding pocket are labeled with green asterisks. Binding sites for the selective agonist UCSF924 are also shown (light blue dot).

The distance between the DRD4 and DRD4rs gene lineages was found to be higher compared to that estimated for the DRD2 gene lineages. In the case of the spotted gar (Lepisosteus oculatus) and the coelacanth (Latimeria chalumnae) the distance, defined as the percentage of amino acid residues that are different between two sequences, was approximately 45% whereas in zebrafish (Danio rerio) it was approximately 49%. The human DRD4 amino acid sequence was aligned to the zebrafish, coelacanth and spotted gar DRD4rs sequence (Fig. 8). The binding sites for dopamine and DRD4 agonists and antagonists are conserved among species. Interestingly, two sites in the hydrophobic pocket of the dopamine receptor differ. The first site is located in the selectivity region of DRD4, where a change from tyrosine (Y) to phenylalanine (F) occurs at position 91 (F91) of the human receptor sequence (Fig. 8; green asterisk). At the second site (Fig. 8; green asterisk) in position 193 of the human DRD4, the isoleucine (I) in the corresponding spotted gar sequence is changed to valine (V) in the other species (V193). To understand the potential effects that these changes might have on DRD4 function we used the recently uncovered crystal structure of the human DRD4 sequence coupled to the antipsychotic drug nemonapride (Wang et al., 2017). All amino acids within 4 Å of the active site are conserved (Figs. 9A and 9B, red amino acids; 9C, red dots) with the exception of the two discussed above. First, F91 located in the recently characterized extended binding pocket, which is a region poorly conserved among dopamine receptors and is key for receptor class specificity (Wang et al., 2017). Second, V193 located in the classic orthosteric-binding pocket known to modulate agonist responses (Lane et al., 2013) (Fig. 9B green amino acids, Fig. 9C, green dots). When we performed molecular dynamics simulation of site-directed mutagenesis to convert the human sequence to the amino acids present in the spotted gar sequence (Fig. 9A, green amino acids), we found that the most likely conformation of Y91 modifies the shape and the ionic properties of the extended binding pocket. Specifically, the polar hydroxyl group oriented along the surface of the pocket would favor interactions with more hydrophilic ligands. Given that the human sequence contains the nonpolar F91 ring, these results could suggest an important evolutionary change in ligand specificity and receptor function. Simulated mutagenesis of V193I, which is located towards the periphery of the orthosteric-binding pocket, slightly modified the shape of the binding pocket; however the nonpolar nature of the amino acid was maintained. Taken together, both substitutions in the human sequence caused the dopamine binding site to be more hydrophobic with less protruding amino acidic side chains, suggesting a structural/functional evolutionary refinement.

Figure 9 Structural details of human DRD4 binding site to the antipsychotic drug nemonapride (in blue) based on the molecular file PDB ID: 5WIV (Wang et al., 2017).

(B) Conserved amino acids within 4 Å of the drug molecule are shown with functional groups (in red). Non-conserved amino acids (in green) were changed (inset A) to the residue present in the fish species: F91Y and V193I. This mutagenesis was simulated choosing the rotamer (orientation of the side chain) with the highest probability (Y rotamer: 72.6%; I rotamer: 79% probability) see methods for additional details. (C) Partial alignment of the human dopamine receptor 4 (DRD4) with zebrafish (Danio rerio), coelacanth (Latimeria chalumnae) and spotted gar (Lepisosteus oculatus) dopamine receptor 4rs (DRD4rs) showing the numbers corresponding to the human DRD4 sequence (NP_000788). Conserved and non-conserved aminoacids shown in (B) are indicated with red and green dots respectively. Non-conserved aminoacids within the region are also shown in green fonts.

Duplicative history and ancestral gene repertoires

To understand the duplicative history of dopamine receptors, including the definition of ancestral repertoires, it is necessary to reconcile the evolutionary history of the gene lineages with the sister group relationships among the species involved. According to our results, the presence of differentiated dopamine receptors in vertebrates (Fig. 5) allowed us to infer that at some point of time the vertebrate ancestor possessed two dopamine receptors, one of each class (Fig. 10). After the two rounds of whole genome duplications (WGD) that occurred in the ancestor of the group (Garcia-Fernàndez & Holland, 1994; Dehal & Boore, 2005) each ancestral gene (DRD1anc and DRD2anc) gave rise to four genes in each class of dopamine receptors (Fig. 10). In support of this hypothesis, the DRD1 and DRD2 classes of dopamine receptors appear in the repository of genes that originated and were retained after the WGDs occurred in the ancestor of vertebrates (Singh, Arora & Isambert, 2015). The fact that non-vertebrate chordates possess just one DRD1 (Kamesh, Aradhyam & Manoj, 2008; Burman et al., 2009) and that the four chromosomal locations where the DRD1 class of receptors are located in humans derive from a single linkage group in the chordate ancestor (Putnam et al., 2008) provide support to our hypothesis. Overall, three out of the four DRD1 originated as a product of the WGDs were retained in the genome of the vertebrate ancestor (DRD1, DRD5 and DRD1C∕E; Fig. 10). After that, in the gnathostome ancestor the DRD1C∕E gene underwent a duplication event that gave rise to the actual DRD1C and DRD1E genes (Fig. 10). In support of this, we recovered a cyclostome sequence sister to the clade containing the DRD1C and DRD1E genes. Thus, the gnathostome ancestor that existed between 615 and 473 mya had a repertoire of four DRD1 genes: DRD1, DRD5, DRD1C and DRD1E (Fig. 10). In teleost fish, a group that experienced an extra round of whole genome duplication (Meyer & Van de Peer, 2005; Kasahara et al., 2007; Sato & Nishida, 2010; Glasauer & Neuhauss, 2014), all DRD1 doubled in number, however, three out of the four gene lineages retained duplicated copies (Fig. 5) (Yamamoto et al., 2013; Yamamoto et al., 2015).

Figure 10 An evolutionary hypothesis regarding the origin of dopamine receptor genes in vertebrates.

The vertebrate ancestor possessed two dopamine receptors, one of each class. However, after the two rounds of whole genome duplications (WGD) that occurred in the ancestor of the group each ancestral gene (DRD1anc and DRD2anc) gave rise to four genes. In the case of the DRD1 class of receptors three out of the four genes were retained in the genome of the vertebrate ancestor. In the gnathostome ancestor, the DRD1C∕E gene underwent a duplication event that gave rise to the actual DRD1C and DRD1E genes. Thus, the gnathostome ancestor had a repertoire of four DRD1 genes: DRD1, DRD5, DRD1C and DRD1E. In the case of the DRD2 group of receptors, the vertebrate WGDs originated four genes, three of which were maintained in the genome of extant species (DRD2∕2l, DRD3 and DRD4∕4rs). In the ancestor of gnathostomes, the DRD2∕2l gene underwent a duplication event that gave rise to an extra copy of the gene. Similarly, the DRD4∕4rs gene also underwent a duplication event that gave rise to an extra copy of the gene. Thus, the ancestor of gnathostome vertebrates possessed a repertoire of five DRD2 genes: DRD2, DRD2l, DRD3, DRD4 and DRD4rs.

Similarly to the DRD1 class of receptor, the vertebrate specific WGDs originated four DRD2 genes, three of which were maintained in the genome of extant species (DRD2∕2l, DRD3 and DRD4∕4rs; Figs. 5 and 10). In the ancestor of gnathostomes the DRD2∕2l gene underwent a duplication event that gave rise to the actual DRD2 and DRD2rs genes (DRD2l; Figs. 4 and 10). In this case both genes followed different evolutionary trajectories. On one hand DRD2 was retained in the genome of all of the main groups of vertebrates (Fig. 5) whereas DRD2l was only retained in coelacanths and bony fish (Fig. 5) (Yamamoto et al., 2015). Similarly, the DRD4∕4rs gene also underwent a duplication event that gave rise to the actual DRD4 and DRD4rs genes (DRD4rs; Figs. 7 and 10). This case is similar to that found for the DRD2 gene, as one of the copies (DRD4) was retained in the genome of all of the main groups of vertebrates, while the other was independently lost in tetrapods and cartilaginous fish (Fig. 6). Consequently, the ancestor of gnathostome vertebrates possessed a repertoire of five DRD2 class of dopamine receptors: DRD2, DRD2l, DRD3, DRD4 and DRD4rs (Fig. 10). As a consequence of the teleost-specific genome duplication (Meyer & Van de Peer, 2005; Kasahara et al., 2007; Sato & Nishida, 2010; Glasauer & Neuhauss, 2014), teleost fish doubled their number of DRD2 receptors, however extant species retained duplicated copies in just two gene lineages (Fig. 5) (Yamamoto et al., 2015).

Concluding Remarks

We present an evolutionary study of the dopamine receptors with special emphasis on unraveling the phylogenetic relationships of the D1 class of receptors and the time of origin of the DRD2l and DRD4rs gene lineages. Our study comprised taxonomic sampling that included representative species of all main groups of vertebrates in addition to other vertebrate biogenic amine receptors. Thus, we were able to reconstruct in a single phylogenetic tree the evolutionary history of both classes of dopamine receptors. In the case of the DRD1 class, our results propose a new phylogenetic hypothesis in which DRD1C was recovered sister to DRD1E and this clade was recovered sister to a cyclostome sequence. DRD1 was recovered sister to the aforementioned clade, and the group containing the DRD5 sequences was sister to all other DRD1 paralogs. According to our phylogenetic tree, the evolutionary origin of the DRD2l and DRD4rs gene lineages would have happened in the ancestor of gnathostomes between 615 and 473 mya, which differs from current proposed scenarios. Of special interest is the analysis of sequences required for dopaminergic neurotransmission. We found high conservation of agonist and antagonist sites suggesting evolutionary conserved dopaminergic pathways. We also found small variation in the dopamine-binding regulatory regions showing a refinement of ligand specificity and big variations in G protein-coupling sequences suggesting differences in downstream signaling cascades through evolution. These new data on evolutionary divergence may help with the rational design of new agonist and antagonist to modulate the dopaminergic pathway.

Supplemental Information

Table S1 Accession numbers of dopamine receptor genes used in this study

Click here for additional data file.

Figure S1 Alignment used for Fig. 1

Click here for additional data file.

Supplemental Information 3 Modified PDB ID 5WIV molecular structure

Modified PDB ID 5WIV file containing the following changes: F91Y and V193I

Click here for additional data file.

Supplemental Information 4 UCSF Chimera

UCSF Chimera file containing the sequence and graphic display of the modified PDB ID 5WIV file as shown in Fig. 9.

Click here for additional data file.

Supplemental Information 5 Best tree Fig. 1

Click here for additional data file.

Supplemental Information 6 AlingX (VNTI express) file

AlingX (VNTI express) file containing the alignment of the sequence files listed below.

Input files PDB ID 5WIV, NP000786 (human DRD2), NP_922917 (Zebrafish DRD2L) NP000788 (human DRD4), NP001012638 (Zebrafish DRD4rs), Coelacanth DRD2 and DRD4, Spotted gar DRD2 and DRD4.

Click here for additional data file.

Additional Information and Declarations

Competing Interests

Author Contributions

Data Availability

The authors declare there are no competing interests.

Juan C. Opazo conceived and designed the experiments, performed the experiments, analyzed the data, contributed reagents/materials/analysis tools, prepared figures and/or tables, authored or reviewed drafts of the paper, approved the final draft.

Kattina Zavala conceived and designed the experiments, performed the experiments, analyzed the data, prepared figures and/or tables, approved the final draft.

Soledad Miranda-Rottmann conceived and designed the experiments, performed the experiments, analyzed the data, prepared figures and/or tables, authored or reviewed drafts of the paper, approved the final draft.

Roberto Araya conceived and designed the experiments, performed the experiments, analyzed the data, contributed reagents/materials/analysis tools, prepared figures and/or tables, authored or reviewed drafts of the paper, approved the final draft.

The following information was supplied regarding data availability:

The data are provided as Supplemental Files.

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
