# Peer review of "Evolution of dopamine receptors: phylogenetic evidence suggests a later origin of the DRD2l and DRD4rs dopamine receptor gene lineages"

_PeerJ, doi:10.7717/peerj.4593_

## Round 0.1 · original submission · Major Revisions

Both reviewers like your manuscript in principle but have some comments that should be addressed. Please note that reviewer 1 submitted their review as a pdf file, which you can obtain by clicking on the "download annotated manuscript" link.

Reviewer 2 ·

Basic reporting

I believe that the data support the conclusions of the study

Experimental design

Not being a evolutionary biologist I cannot judge of the experimental design

Validity of the findings

As far as I understand the data are acceptable. I only find that from an outside point of view little discovery is revealed in this manuscript

Additional comments

The manuscript is well written. I cannot judge of all the data but they seem to be expected. I only would like that the authors define what is D2l and D4rs. This is difficult for pharmacologist as D2l relates to the long form of the D2 receptor and D4rs is reminiscent of the different D4 subtypes. Also the comments on the role of the D4 in psychiatric disorders should be toned down as thus far the D4 receptor has not been proven to be associated with any disorder

---

## Round 0.2 · Minor Revisions

Both reviewers appreciated the improvements you made. However, reviewer 1 still has some concerns about details of the methods. I agree with the comments of reviewer 1, in particular with respect to the availability of any code used to generate the results.

·

Basic reporting

N/A

Experimental design

N/A

Validity of the findings

N/A

Additional comments

The authors have addressed my comments, but I think a bit more detail still needed, specifically:

1. For the BLAST search performed, which sequence was used as the seed sequence? Which BLAST algorithm was run? How was "conserved synteny" identified? How were sequences manually identified if a software was also used for the same step? The details here are improved but are still too thin. In addition, any code written and employed for these tasks MUST be released with the paper either as supplementary information or on a linked github repository, or similar.

2. Running tree inference "multiple times" is not a specific number. Is this 5 times? 10 times? 100 times?You need to give a real value and as needed infer the tree again so that you can give a specific value.

Reviewer 2 ·

Basic reporting

The manuscript is acceptable

Experimental design

No issue

Validity of the findings

Acceptable

Additional comments

You mention the variability in the DRD receptor nomenclature as stemming from the fact that these receptors were originally not looked from a phylogenetic standpoint. DRDs are most important in neuropharmacology and as such are discussed as pharmacological entities by most.

---

## Round 0.3 · Minor Revisions

I looked over your revisions and found that many of the methods were still described in insufficient detail. To get a second opinion, I ran your revised manuscript by Reviewer 1, who came to the same conclusion. In general, as currently presented, your work is not reproducible, because methods are described in broad general terms rather than with specific details. You're using several different softwares, all of which have various parameter settings or configuration options, and you're not providing any details about these settings or configurations. As an example of what I mean by providing specific details, take a look at this paper from my own group: https://peerj.com/articles/3391/#methods

·

Basic reporting

N/A

Experimental design

N/A

Validity of the findings

N/A

Additional comments

I feel some clarification is still needed, in particular:

1. My comment regarding this sentence was not answered: "Dopamine receptor genes were manually annotated by comparing known exon sequences to genomic pieces using the program Blast2seq (Tatusova and Madden 1999)."

Please explain how you can manually annotate while using a program - what is the role of this program vs your manual annotation?

2. The updated sentence "Phylogenetic analyses were 20 times in order to better explore the tree space." needs a verb, i.e. "were PERFORMED 20 times".

3. The level of explanation regarding "synteny analysis" is still far too thin. It does not sound as though any synteny analysis was performed, just that the authors read a paper which claimed to identify synteny. The authors MUST be PRECISE about their analysis here.

4. The authors may not have written code, but they have certainly used software for which specific parameters must be specified. The authors must therefore include the following in the manuscript:
- specific *parameters* used in the tblastn search (i.e. evalue, database searched, etc.)
- specific arguments, and all input/output files generated, for the molecular dynamics simulations
- details with this sentence: "Sequences were aligned using Vector NTI 169 Express (Thermo Fisher)". What alignment algorithm is used in this program? What arguments were specified?

---

## Round 0.4 · accepted · Accept

Thank you for making these final revisions.